# Non-ohmic tissue conduction in cardiac electrophysiology: Upscaling the non-linear voltage-dependent conductance of gap junctions

Daniel E. Hurtado[1,2,3]*, Javiera Jilberto[1,3], Grigory Panasenko[4,5,6]

**1** Institute for Biological and Medical Engineering, Schools of Engineering, Medicine and Biological Sciences, Pontificia Universidad Católica de Chile, Santiago, Chile, **2** Department of Structural and Geotechnical Engineering, School of Engineering, Pontificia Universidad Católica de Chile, Santiago, Chile, **3** Millennium Nucleus for Cardiovascular Magnetic Resonance, Chile, **4** Institute Camille Jordan, Université Jean Monnet, Université de Lyon, Saint-Etienne, France, **5** Institute of Applied Mathematics, Vilnius University, Vilnius, Lithuania, **6** National Research University "Moscow Power Engineering Institute", Moscow, Russia

* dhurtado@ing.puc.cl

**Data Availability Statement:** All codes that generated the data included in the paper are freely available at the GitHub repository https://github.com/dehurtado/NonOhmicConduction.

## Abstract

Gap junctions are key mediators of intercellular communication in cardiac tissue, and their function is vital to sustaining normal cardiac electrical activity. Conduction through gap junctions strongly depends on the hemichannel arrangement and transjunctional voltage, rendering the intercellular conductance highly non-Ohmic, particularly under steady-state regimes of conduction. Despite this marked non-linear behavior, current tissue-level models of cardiac conduction are rooted in the assumption that gap-junctions conductance is constant (Ohmic), which results in inaccurate predictions of electrical propagation, particularly in the low junctional-coupling regime observed under pathological conditions. In this work, we present a novel non-Ohmic homogenization model (NOHM) of cardiac conduction that is suitable to tissue-scale simulations. Using non-linear homogenization theory, we develop a conductivity model that seamlessly upscales the voltage-dependent conductance of gap junctions, without the need of explicitly modeling gap junctions. The NOHM model allows for the simulation of electrical propagation in tissue-level cardiac domains that accurately resemble that of cell-based microscopic models for a wide range of junctional coupling scenarios, recovering key conduction features at a fraction of the computational complexity. A unique feature of the NOHM model is the possibility of upscaling the response of non-symmetric gap-junction conductance distributions, which result in conduction velocities that strongly depend on the direction of propagation, thus allowing to model the normal and retrograde conduction observed in certain regions of the heart. We envision that the NOHM model will enable organ-level simulations that are informed by sub- and inter-cellular mechanisms, delivering an accurate and predictive in-silico tool for understanding the heart function. Codes are available for download at https://github.com/dehurtado/NonOhmicConduction.

**Funding:** DEH acknowledges the support from CONICYT thorugh grant FONDECYT Regular 1180832. DEH and JJ received funding from Millennium Science Initiative of the Ministry of Economy, Development and Tourism of Chile, grant Nucleus for Cardiovascular Magnetic Resonance. GP received funding from Russian Science Foundation Grant #19-11-00033. The funders had no role in study design, data collection and analysis, decision to publish, or preparation of the manuscript.

**Competing interests:** The authors have declared that no competing interests exist.

## Author summary

The heart relies on the propagation of electrical impulses that are mediated gap junctions, whose conduction properties vary depending on the transjunctional voltage. Despite this non-linear feature, current mathematical models assume that cardiac tissue behaves like an Ohmic (linear) material, thus delivering inaccurate results when simulated in a computer. Here we present a novel mathematical multiscale model that explicitly includes the non-Ohmic response of gap junctions in its predictions. Our results show that the proposed model recovers important conduction features modulated by gap junctions at a fraction of the computational complexity. This contribution represents an important step towards constructing computer models of a whole heart that can predict organ-level behavior in reasonable computing times.

## Introduction

The conduction of electrical waves in cardiac tissue is key to human life, as the synchronized contraction of the cardiac muscle is controlled by electrical impulses that travel in a coordinated manner throughout the heart chambers. Under pathological conditions cardiac conduction can be severely reduced, potentially leading to reentrant arrhythmias and ultimately death if normal propagation is not restored properly [1]. At a subcellular level, electrical communication in cardiac tissue occurs by means of a rapid flow of ions moving through the cytoplasm of cardiac cells, and a slower intercellular flow mediated by gap junctions embedded in the intercalated discs. Gap junctions are intercellular channels composed by hemichannels of specialized proteins, known as connexins, that control the passage of ions between neighboring cells [2]. The regulation of ionic flow through gap junctions has been established for a variety of connexin types and hexameric arrangements, which under dynamic conditions result in a markedly non-linear relation between the electric conductance and the transjunctional voltage [3], revealing a non-ohmic electrical behavior. Further, it has been shown that ionic flow through cell junctions can take up to 50% of the total conduction time in cultured strands of myocytes with normal coupling levels [4], and that conduction velocity (CV) is largely controlled by the level of gap-junctional communication [1, 5], which highlights the key physiological relevance of gap-junction conductivity and coupling in tissue electrical conduction.

Cardiac modeling and simulation has strongly motivated the development of tissue-level mathematical models of electrophysiology, as they have the ability to connect subcellular mechanisms to whole-organ behavior [6]. To date, the vast majority of continuum models assume a linear conduction model of spatial communication, based on the assumption that electrical current in cardiac tissue follows Ohm's law, i.e, that current is linearly proportional to gradients in the intra-cellular potential [7, 8]. From a mathematical perspective, the assumption that conduction in cardiac tissue follows Ohm's law is conveniently represented by a linear diffusion term, where gradients are modulated by a conductivity tensor that is independent of the local electrical activity. The most complete electrophysiology formulation is given by the bidomain model [9] where both the intra-cellular and extra-cellular potential fields are considered. Further, by assuming that the intra- and extra-cellular conductivity tensors have the same anisotropy ratio, the bidomain equations can be conveniently represented by a non-linear reaction-diffusion partial differential equation known as the monodomain (cable) model [10].

Using two-scale asymptotic homogenization techniques, analytic expressions have been obtained for the effective conductivity tensor, which is then used to model the electrical current in an average macroscopic sense [11–13]. To this end, periodicity at the microstructural level of cardiac tissue is assumed, and a representative tissue unit is partitioned in regions of high and low conductivity that represent the cytoplasm and intercalated discs with gap junctions, respectively. While this approach allows for the explicit consideration of regions with decreased conductivity, e.g. membranes where flow is mediated by gap junctions, Ohm's law is still assumed to hold throughout the microstructural domain [14]. As a result, the non-Ohmic behavior of gap junctions and their impact on tissue-level conduction continues to be neglected [15]. In particular, it has been shown that continuum models that consider effective conductivity tensors described above fail to capture the slow conduction of electrical impulses in cases of low gap-junctional coupling [13, 16], limiting their applicability to the simulation of pathological conditions in excitable tissue. Non-linear diffusion models that replace the conduction term in the monodomain equation either by a fractional laplacian [17, 18] or a porous-medium-like diffusive term [8, 19] have been recently proposed. While these formulations have shown to modulate the shape of propagating waves and other restitution properties, they remain largely phenomenological, and have the disadvantage of not being able to upscale microscopic physical information, neither have been assessed for cases of low junctional coupling.

In this work, we present a multiscale continuum model of cardiac tissue conduction that accounts for the nonlinear communication between adjacent cells. We argue that the explicit consideration of the non-ohmic behavior of gap junctions can be seamlessly embedded into continuum tissue-scale models of electrophysiology using an asymptotic homogenization approach, which delivers nonlinear continuum equations for characterizing the electrical conduction in excitable media.

## Methods

### Multiscale tissue model for non-ohmic conduction

In the following we consider the microscopic problem of non-linear conduction in a strand of cardiac cells with domain $\Omega = (0, L)$, see Fig 1. We let $\varepsilon$ be the cell length, $\delta\varepsilon$ be the length of gap junctions, and assume that $\delta\varepsilon \ll \varepsilon \ll L$. Further, we let $u_{\varepsilon,\delta}$ be the microscopic transmembrane potential field, and $j_{\varepsilon,\delta}$ be the microscopic current density. The time-independent problem of conduction resulting from current balance reads

$$-\frac{\partial}{\partial x} j_{\varepsilon,\delta}(u_{\varepsilon,\delta}) = 0, \quad x \in \Omega, \tag{1}$$

and we note that in writing (1) it has been assumed that the extra-cellular potential is constant along the cardiac strand. We denote the space occupied by the cytoplasm by $B_{\varepsilon,\delta}^{cyt} = \cup_{k=-\infty}^{\infty} \left( \left(k + \frac{\delta}{2}\right)\varepsilon, \left(k + 1 - \frac{\delta}{2}\right)\varepsilon \right)$, and the space occupied by gap junctions by $B_{\varepsilon,\delta}^{gap} = \cup_{k=-\infty}^{\infty} \left( \left(k - \frac{\delta}{2}\right)\varepsilon, \left(k + \frac{\delta}{2}\right)\varepsilon \right)$. Further, we assume that current is governed by Ohm's law inside the cytoplasm with conductivity $\sigma_c$, but is non-linearly regulated at the gap junctions, which we express by the following microscopic constitutive law

$$j_{\varepsilon,\delta}(u_{\varepsilon,\delta}) = -\sigma\big(x, \{u_{\varepsilon,\delta}\}\big) \frac{\partial u_{\varepsilon,\delta}}{\partial x} \tag{2}$$

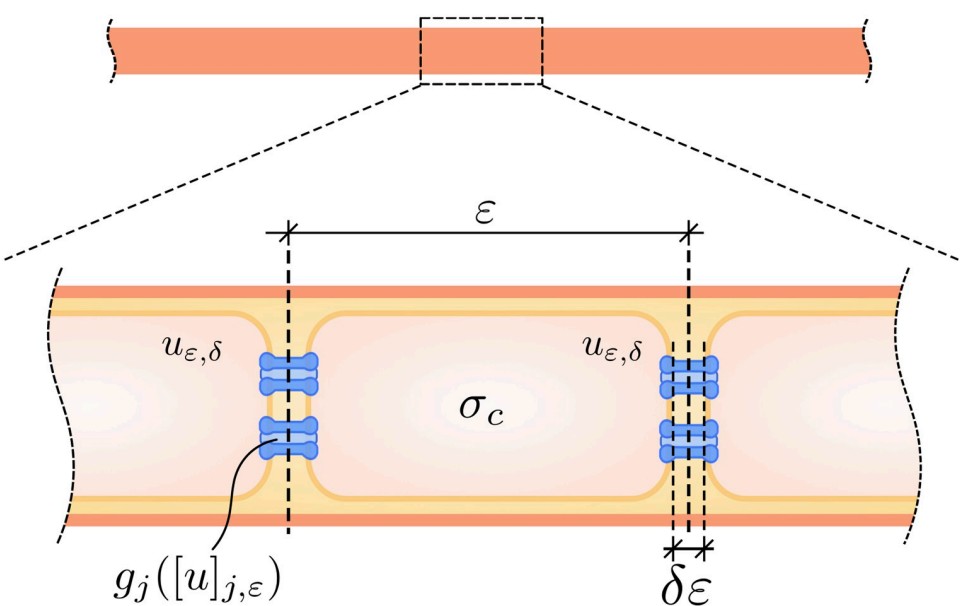

**Fig 1. Schematic of the multiscale model of cardiac conduction.** Ionic currents are linearly proportional to gradients of transmembrane potential inside the cytoplasm, but are non-linearly mediated by gap junctions located at the intercalated discs.

where the conductivity is described by the following relation

$$\sigma(x,u) = \begin{cases} \sigma_c, & x \in B^{cyt}_{\varepsilon,\delta}, \\[2em] \delta\sigma_g(1 + \mu a(S\,[u]_{j,\varepsilon})), & x \in B^{gap}_{\varepsilon,\delta}, \end{cases} \tag{3}$$

where $\delta\sigma_g$ is a representative conductivity for the intercalated disc with gap junctions, $\mu$ is a positive constant, and $a$ is a smooth bounded function that depends on the transjunctional voltage jump defined as

$$[u]_{j,\varepsilon} = u((k + \tfrac{\delta}{2})\varepsilon) - u((k - \tfrac{\delta}{2}),\varepsilon) \tag{4}$$

where $S$ is a scaling parameter ($S \le \varepsilon^{-1}$). From (3) and the relationship between electrical conductance and conductivity for a cylindrical domain we write

$$\delta\sigma_g(1 + \mu a(S\,[u]_{j,\varepsilon})) = \beta\frac{g_{jo}\delta\varepsilon}{A_{cell}}g_j(S\,[u]_{j,\varepsilon}), \tag{5}$$

where $g_{jo}$ is a representative conductance for the intercalated disc with gap junctions, $A_{cell}$ the cross-sectional area of the cell, and the parameter $\beta$ represents the level of gap-junctional coupling (GJc), with $\beta \in [0, 1]$. From (5), we set $\delta\sigma_g = \frac{g_{jo}\delta\varepsilon}{A_{cell}}$. As a result, we get

$$\mu a(S\,[u]_{j,\varepsilon}) = \beta g_j(S[u]_{j,\varepsilon}) - 1. \tag{6}$$

Using asymptotic analysis (see S1 Appendix for technical details and proofs) we show that the macroscopic (tissue-level) current conservation for the steady-state problem is governed

by the homogenized equation

$$\frac{\partial}{\partial x}\left(\hat{\sigma}\left(\frac{dv}{dx}\right)\frac{\partial v}{\partial x}\right) = 0, \quad x \in \Omega,$$

(7)

where $v$ is the macroscopic transmembrane potential, and the effective conductivity modulating conduction at the macroscopic scale takes the form

$$\hat{\sigma}(y) = \sigma_c \left\{ \frac{1 + \mu a(S\varepsilon[N](y))}{\frac{\sigma_c}{\sigma_g} + (1-\delta)(1 + \mu a(S\varepsilon[N](y)))} \right\}$$

(8)

where

$$[N](y) = -(1-\delta)\left(\frac{\hat{\sigma}(y)}{\sigma_c} - 1\right)y,$$

(9)

with $[N] = [u]_{j,\varepsilon}/\varepsilon$, and we note that for a given transmembrane potential gradient $y$, the effective conductivity $\hat{\sigma}(y)$ is implicitly solved from (8) and (9). Further, we show that under reasonable assumptions, the following error estimate for the macroscopic transmembrane potential holds

$$\|u_{\varepsilon,\delta} - v\|_{L^\infty((0,1))} = O(\varepsilon + \delta^2).$$

(10)

We now focus on the time-dependent macroscopic model of cardiac electrophysiology for the time interval $(0, T)$. The homogenized electrical flux described in the right-hand side of (7) is then balanced by the transmembrane current, leading to the non-Ohmic cable equation

$$\frac{\partial}{\partial x}\left(\hat{\sigma}\left(\frac{dv}{dx}\right)\frac{\partial v}{\partial x}\right) = A_m\left\{C_m\frac{\partial v}{\partial t} + I_{ion}\right\} \quad \text{in } \Omega \times (0, T),$$

(11)

where $I_{ion} : \mathbb{R} \times \mathbb{R}^M \to \mathbb{R}$ represents the transmembrane ionic current, $C_m$ is the membrane capacity and $A_m$ is the surface-to-volume ratio, and we note that the right-hand side of (11) accounts for the amount of charge that leaves the intra-cellular domain and enters the extra-cellular domain. Further, we will assume that the transmembrane ionic current $I_{ion}$ is governed by $v$ and by gating variables $w : \Omega \times (0, T) \to \mathbb{R}^M$ that modulate the conductance of ion channels, pumps and exchangers, i.e., $I_{ion} = I_{ion}(v, w)$, where the exact functional form of $I_{ion}$ will depend on the choice of ionic model. The evolution of gating variables is determined by kinetic equations of the form

$$\frac{\partial w}{\partial t} = g(v, w),$$

(12)

where the form of $g : \mathbb{R} \times \mathbb{R}^M \to \mathbb{R}^M$ will also depend on chosen the ionic model. The Eqs (11) and (12) are supplemented with initial and boundary conditions for the transmembrane potential and gating variables to form an initial boundary value problem, which we refer to as the Non-Ohmic Homogenization Model (NOHM). Boundary conditions at the left end prescribed a pulsatile electrical current, while the right end was prescribed with zero current. To study reverse conduction, the boundary conditions where flipped. The numerical solution of the coupled system of the non-Ohmic cable Eq (11) and kinetic Eq (12) was performed using a standard Galerkin finite-element scheme [20] for the spatial discretization and a Forward Euler scheme for the time discretization implemented in FEniCS [21], see S1 Appendix. Codes are available for download at https://github.com/dehurtado/NonOhmicConduction.

For the sake of comparison, we also consider the case of uniform (Ohmic) gap junction conductivity, i.e.,

$$\sigma(x, u) = \begin{cases} \sigma_c, & x \in B_{\varepsilon,\delta}^{cyt}, \\ \\ \beta\delta\sigma_g, & x \in B_{\varepsilon,\delta}^{gap}. \end{cases} \tag{13}$$

Following standard asymptotic-analysis arguments for linear systems [11], one can show that for the case of the piecewise uniform conductivity tensor defined in (13) the effective conductivity tensor takes the form

$$\hat{\sigma} = \left\{ \frac{1}{\sigma_c} + \frac{1}{\beta\sigma_g} \right\}^{-1}. \tag{14}$$

We remark that in this case the macroscopic conductivity is not dependent on the voltage gradient. Further, we note that (14) is also obtained as a particular case of the NOHM when the microscopic conductivity (13) is assumed, see S1 Appendix. We refer to the system of Eqs (11) and (12) that considers the uniform effective conductivity tensor (14) as the Linear Homogenization Model (LHM).

## Cellular models of cardiac propagation

To validate the proposed NOHM model we consider the cellular model described in [7, 22], in which a strand of cardiac cells electrically connected by gap junctions are represented using a circuit network, see Fig 2. In the following, we summarize the main aspects of cellular modeling. For the cellular models (CM), a chain of cells is discretized at subcellular level in $n_{div} = 10$ subregions. A generic node $i$ is connected to its neighbor $i + 1$ through a resistor $R^{i,i+1}$. If $i$ and $i + 1$ belongs to the same cell $R^{i,i+1} = R_{myo}$ where $R_{myo}$ is the myoplasmic resistance. Now, if $R^{i,i+1}$ connects nodes from different cells (i.e. a gap junction) its value will be given by $R^{i,i+1} = R_j(V_j)$ where $R_j(V_j)$ is the resistance of the gap junction, which depends non-linearly on the transjunctional voltage $V_j$. For computing the resistance $R^{i,i+1}$ we use

$$V_j = v^{i+1+(ndiv-1)} - v^{i-(ndiv-1)}$$

where we consider how the experimental relationship $V_j$ versus $g_j$ is obtained through the dual voltage clamp method [23].

Due to the Kirchhoff's law, the current balance yields to the following finite-difference equation

$$\sigma^{i,i+1} \frac{v^{i+1} - v^i}{\Delta x^2} + \sigma^{i-1,i} \frac{v^{i-1} - v^i}{\Delta x^2} = A_m \left\{ C_m \frac{\partial v^i}{\partial t} + I_{ion}(v^i, w^i) \right\}, \tag{15}$$

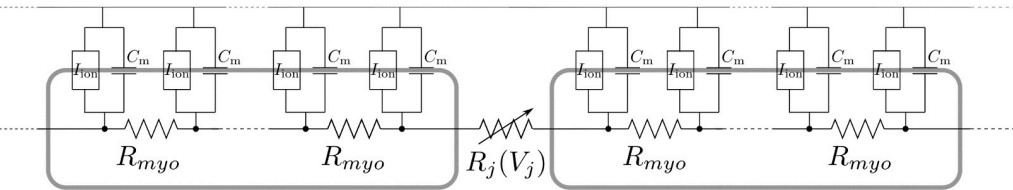

**Fig 2. Circuit representation for the cellular models.**

where $A_m$ is the surface to volume ratio and $C_m$ the membrane capacitance. The ionic current $I_{ion}$ depends on the transmembrane potential in the node $v^i$ and on the gating variables $w^i$. Local conductivity can be expressed in terms of the resistance as

$$\sigma^{i,i+1} = \frac{\Delta x}{R^{i,i+1} A_{cell}}$$

with $\Delta x = l_{cell}/n_{div}$. Then

$$\sigma^{i,i+1} = \begin{cases} \sigma_c & \text{if } i, i+1 \text{ are in the same cell} \\ \delta\sigma_g \beta g_j(V_j) & \text{if } i, i+1 \text{ are in different cells} \end{cases} \tag{16}$$

and we call the model given by Eqs (15) and (16) CM voltage-gated. If the gap junctions are assumed to be voltage insensitive, then the conductivity reads

$$\sigma^{i,i+1} = \begin{cases} \sigma_c & \text{if } i, i+1 \text{ are in the same cell,} \\ \beta\delta\sigma_g & \text{if } i, i+1 \text{ are in different cells.} \end{cases} \tag{17}$$

We refer to the model given by Eqs (15) and (17) as the CM clamped, and we note that it serves as a cellular counterpart to the LHM.

## Conduction experiments in a cardiac strand

In this work, we model the propagation of electrical impulses in a strand with a length $L = 6.4$ mm. Cells were assumed to be cylinders with radius $r_{cell} = 11 \ \mu m$, length $\varepsilon = 100 \ \mu$m, and intercalated-disc length ratio of $\delta = 10^{-4}$. Based on these dimensions, $A_{cell} = 380 \ \mu$m$^2$. For the gap-junction conductance model, we set $S = 2$, and assumed $g_{jo} = 2.534 \ \mu$m [24], which results in a representative conductivity of $\delta\sigma_g = 6.67 \cdot 10^{-5}$ Sm$^{-1}$. The cytoplasmic conductivity and the membrane capacitance are taken to be $\sigma_c = 0.667$ Sm$^{-1}$ and $C_m = 1 \ \mu$F/cm$^2$, respectively [24]. For the transmembrane ionic current, we considered the Luo-Rudy I model [25]. The surface to volume ratio is given by $A_m = 2RCG/r_{cell}$, where RCG = 2 is the ratio between capacitive and geometrical areas and [24, 26]. Given the time-dependent behavior of the conductance of GJs [5, 27], in our experiments, we consider two limit cases for the temporal state of gap junctions: the *instantaneous conductance* case, and the *steady-state conductance* case. For the instantaneous conductance case, we adopt the model of Vogel and Weingart [28] which for the normalized conductance takes the form

$$g_{j,inst}(V_j) = \begin{cases} \dfrac{G_j^-}{e^{\frac{-V_j}{V_H^-}\left(1+e^{V_j/V_H^-}\right)} + e^{\frac{V_j}{V_H^-}\left(1+e^{-V_j/V_H^-}\right)}} & V_j < 0, \\[4em] \dfrac{G_j^+}{e^{\frac{-V_j}{V_H^+}(1+e^{V_j/V_H^+})} + e^{\frac{V_j}{V_H^+}(1+e^{-V_j/V_H^+})}} & V_j > 0, \end{cases} \tag{18}$$

where $V_j$ is the transjunctional voltage, and $G_j^+$, $G_j^-$, $V_H^+$, $V_H^-$ are parameters. For the steady-state case, we assume that the normalized gap-junction conductance follows a Boltzmann

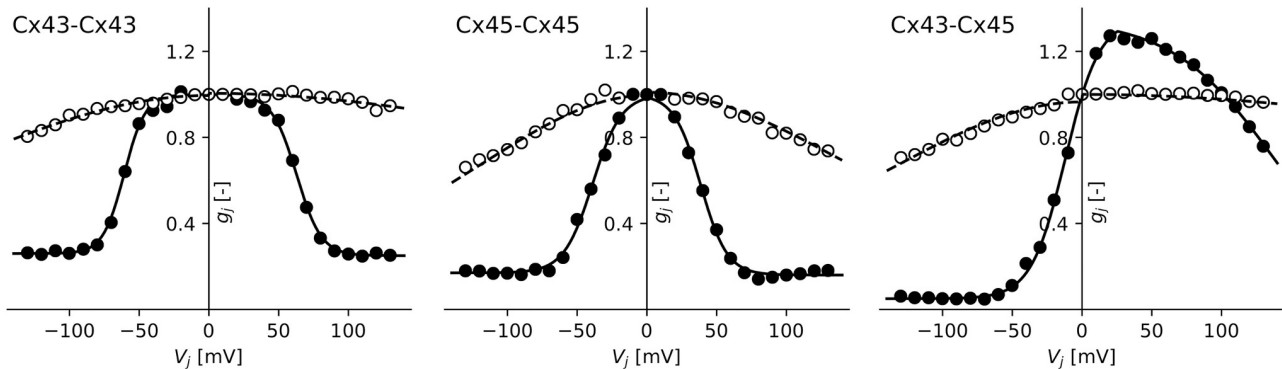

**Fig 3. Normalized conductance of gap junctions as a function of the transjunctional voltage.** (left) Cx43-Cx43 channel, (center) Cx45-Cx45 channel, and (right) Cx43-Cx45 channel. Data extracted from [3]. The (○) and (•) data corresponds to instantaneous and steady state conductance, respectively.

distribution [29] that reads

$$
g_{j,ss}(V_j) = \begin{cases} \dfrac{1-g_{j,min}^{+}}{p+e^{(A^{+}(V_j-V_{j0}^{+}))}} + g_{j,min}^{+} & V_j < d, \\[2ex] \dfrac{1-g_{j,min}^{-}}{p+e^{(A^{-}(V_j-V_{j0}^{-}))}} + g_{j,min}^{-} & V_j > d, \end{cases}
\tag{19}
$$

where $p, d, g_{j,min}^{+}, g_{j,min}^{-}, A^{+}, A^{-}, V_{j0}^{+}, V_{j0}^{-}$ are parameters that depend on the type of channel.

To assess the performance of the NOHM model under different conductance distributions, we considered three types of gap-junction channels: the homomeric-homotypic channels Cx43-Cx43 and Cx45-Cx45, and the homomeric-heterotypic channel Cx43-Cx45. The normalized conductance distributions for the instantaneous and steady-state cases of these channels are depicted in Fig 3. The parameters for the instantaneous and steady-state conductance models have been reported in the literature [3], and are summarized in Table 1. The parameter $A$ is computed from $z$ as $A = z/kT$, where $kT = 25.7$ meV is the product of the Boltzmann constant $k$ with the temperature $T$. The effect of gap-junctional coupling on conduction is studied by modulating the maximal gap-junction conductance from $\beta = 100\%$ to $\beta = 0.5\%$. The cardiac strand is excited on one end with an applied transmembrane current using a pacing cycle length of 800 ms and whose amplitudes varied between 10 $\mu$A/mm$^2$ to 35 $\mu$A/mm$^2$, which elicits a propagating pulse from left to right. Retrograde-conduction effects are studied by propagating pulses from right to left. Pacing rate dependence was studied by constructing CV restitution curves, using the pacing protocol reported by Gizzi and co-workers [30].

**Table 1. Parameters for the conductance distribution of gap junctions, taken from [3].** For $V_{j0}$, $g_{j,min}$, $z$ the negative/positive values are presented. The Cx43-Cx45 case considered a modified Boltzmann distribution to improve the fitness to data.

| | Instantaneous model | | Steady-state model | | | | |
|---|---|---|---|---|---|---|---|
| **Channel** | $G_j$ | $V_H$ [mV] | $V_{j0}$ [mV] | $g_{j,min}$ | $z$ | $p$ | $d$ |
| Cx43-Cx43 | 1.99/2.01 | -175.8/318.4 | -60.8/62.9 | 0.26/0.25 | -3.4/2.9 | 1 | 0 |
| Cx45-Cx45 | 1.99/2.02 | -112.7/135.0 | -38.9/38.5 | 0.16/0.17 | -2.5/2.7 | 1 | 0 |
| Cx43-Cx45 | 1.93/2.0 | -130.0/404.0 | -15.9/149.3 | 0.05/0.05 | -2.1/0.7 | 0.73 | 25 |

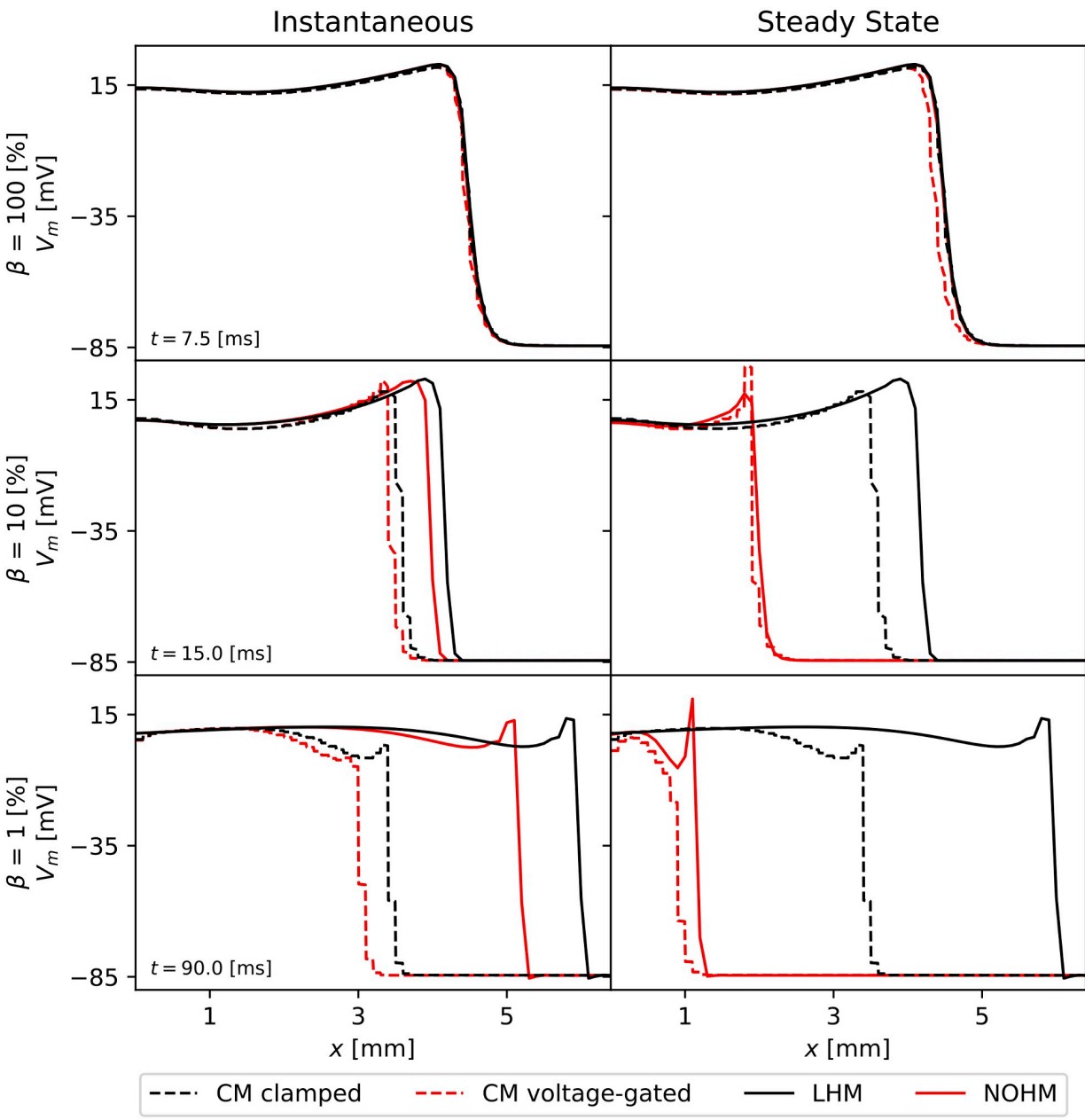

**Fig 4. Impulse conduction features from computational simulations.** The propagating wavefront predicted by the CM clamped, CM voltage-gated, LHM and NOHM are compared for three levels of transjunctional coupling: (top row) high coupling $\beta = 100\%$, (middle row) low coupling $\beta = 10\%$, and (bottom row) very low couping $\beta = 1\%$. In general, the LHM and NOHM drift ahead of their CM counterparts as the GJc is decreased in the case of instantaneous conductance. In contrast, for the case of steady-state conductance the NOHM accurately predicts the CM voltage-gated even for very low coupling levels, whereas the LHM substantially drifts ahead from the CM clamped wavefront.

## Results

The numerical solution of CM clamped, and CM voltage-gated resulted in dynamical systems with 642 degrees of freedom, using cell subdomains with a length of 10 $\mu$m. In contrast, the continuum LHM and the NOHM employed only 65 degrees of freedom, equivalent to a spatial discretization of 100 $\mu$m. Fig 4 shows the propagating wavefronts as predicted by the four conduction models studied in this work for the instantaneous conduction and steady-state

conduction cases of the Cx43-Cx43 channel. For high GJc, $\beta = 100\%$, we observe that all four models predicted a very similar wavefront both for the instantaneous and for the steady-state regimes of conductance (Fig 4, top row). When GJc was reduced to low coupling levels, $\beta = 10\%$, continuum models (LHM and NOHM) resulted in propagating waves that drifted ahead of their CM counterparts (CM clamped and CM voltage-gated) in the case of instantaneous conductance. Interestingly, for the case of steady-state conductance, the NOHM accurately predicted the response of the CM voltage-gated, whereas the LHM drifted ahead of the CM clamped (Fig 4, middle row). Remarkably, for very low GJc, $\beta = 1\%$, the NOHM still delivered an accurate wavefront when compared to the CM voltage-gated in the steady-state conductance case, whereas considerable drift occurred in the instantaneous conductance case both for the LHM and the NOHM (Fig 4, bottom row).

Fig 5 shows the CV as a function of the GJc, measured in terms of $\beta$, for the instantaneous and steady-state conduction cases for homomeric-homotypic channels Cx43-Cx43 and Cx45-Cx45. Under instantaneous conductance (Fig 5, left column) all four models converged to CV values between 64–65 cm/s for the high-coupling case $\beta = 100\%$. Cellular models delivered, in general, a very similar behavior. As GJc was reduced, both LHM and NOHM overestimated the CV when compared to CM clamped and CM voltage-gated, respectively. For the case of steady-state conductance (Fig 5, right column), the CM clamped consistently delivered higher values of CV than the CM voltage-gated. Further, the LHM continued to overestimate the CV when compared to the CM clamped as GJc was reduced. Remarkably, the NOHM closely followed the behavior of the CM voltage-clamped, even for very low GJc. Cellular models predicted conduction block for $\beta \leq 0.5\%$ in all cases. Conduction block was captured by the NOHM for the steady-state conductance regime, but not for the instantaneous case. LHM did not predict the conduction block for the range of $\beta$ analyzed. Retrograde wave propagation experiments for Cx43-Cx43 and Cx45-Cx45 channels (not shown in the figure) resulted in CV curves that did not differ from those obtained in normal wave propagation.

Fig 6 shows the dependence of CV on the GJc for the Cx43-Cx45 channel under normal wave propagation (left-to-right direction) and retrograde wave propagation (right-to-left direction). For the instantaneous conductance case (Fig 6, left column), the CM voltage-gated predicted CVs that were slightly higher in retrograde propagation than in normal propagation. Continuum models overestimated the CV when compared to their cellular counterparts. While the LHM predictions were insensitive to the direction of propagation, the NOHM predicted higher CVs for the case of retrograde wave propagation. For the steady-state conductance case (Fig 6, right column), the CM voltage-gated resulted in CVs that were considerably higher in retrograde propagation than in normal propagation, a feature not captured by the CM clamped, which was insensitive to the direction of propagation. The NOHM was able to capture such large differences in CV as well as the conduction block at $\beta = 10\%$, whereas the LHM only delivered reasonable predictions for the retrograde propagation, but considerably overestimated CVs in the normal propagation case, and did not reach conduction block for any of the $\beta$ values analyzed.

The dependence of CV on the pacing rate and propagation orientation in voltage-dependent models of conduction was studied by constructing CV restitution curves for the case of channel Cx43-Cx45, see Fig 7. In all cases, the CM voltage-gate under retrograde propagation resulted in higher CVs than in normal propagation. The NOHM captured this orientation dependence and delivered CV curves with a similar shape but with higher values for the cases of low and very low GJc. For the case of steady-state conductance, the NOHM also captured the conduction block predicted by the CM voltage-gate for low and very low GJc. The cycle length in all simulations was reduced until loss of propagation, which occurred in the range of 310-330 ms.

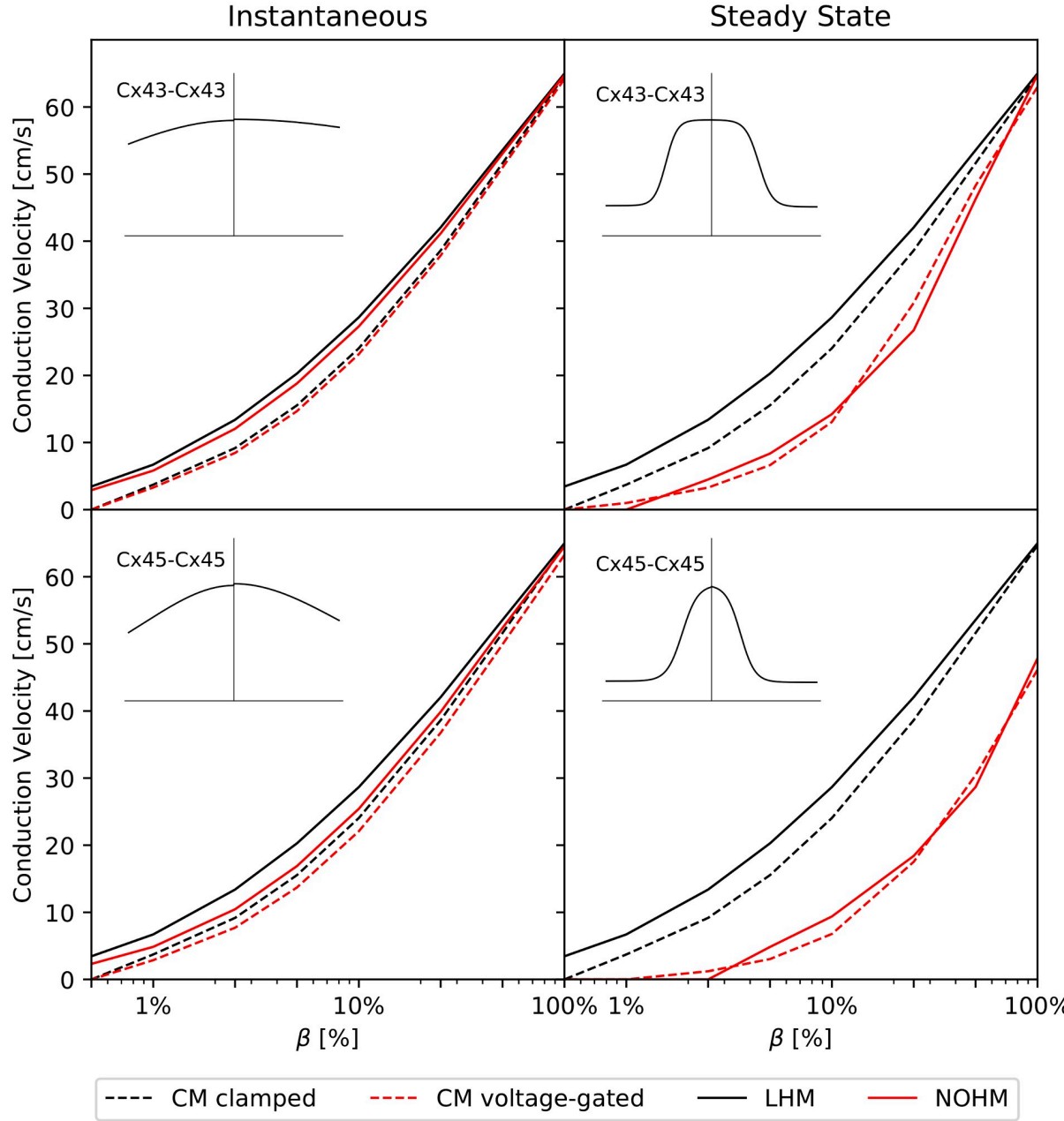

**Fig 5. Conduction velocity studies on a cardiac strand and the effect of gap-junction coupling for symmetric conductance distributions: (top row) Cx43-Cx43 channel, (bottom row) Cx45-Cx45 channel, (left column) instantaneous conductance, (right column) steady-state conductance.** Black and red colors are used to indicate voltage-independent and voltage-dependent gap-junction conduction, respectively. Voltage-dependent models delivered lower conduction velocities than voltage-independent models of gap-junction conductance, particularly for the steady-state regime.

The influence of the spatial discretization on the CV is reported in Fig 8. Mesh sizes, interpreted as cell segments in cellular modes, ranging from $\Delta x = 0.01$ mm to $\Delta x = 0.2$ mm were considered for both the instantaneous and steady-state conduction cases of a Cx43-Cx43 channel under high GJc. In both cases, a strong dependence of the CV on the mesh size was observed for the CM clamped and CM voltage-clamped, with higher CVs for larger mesh

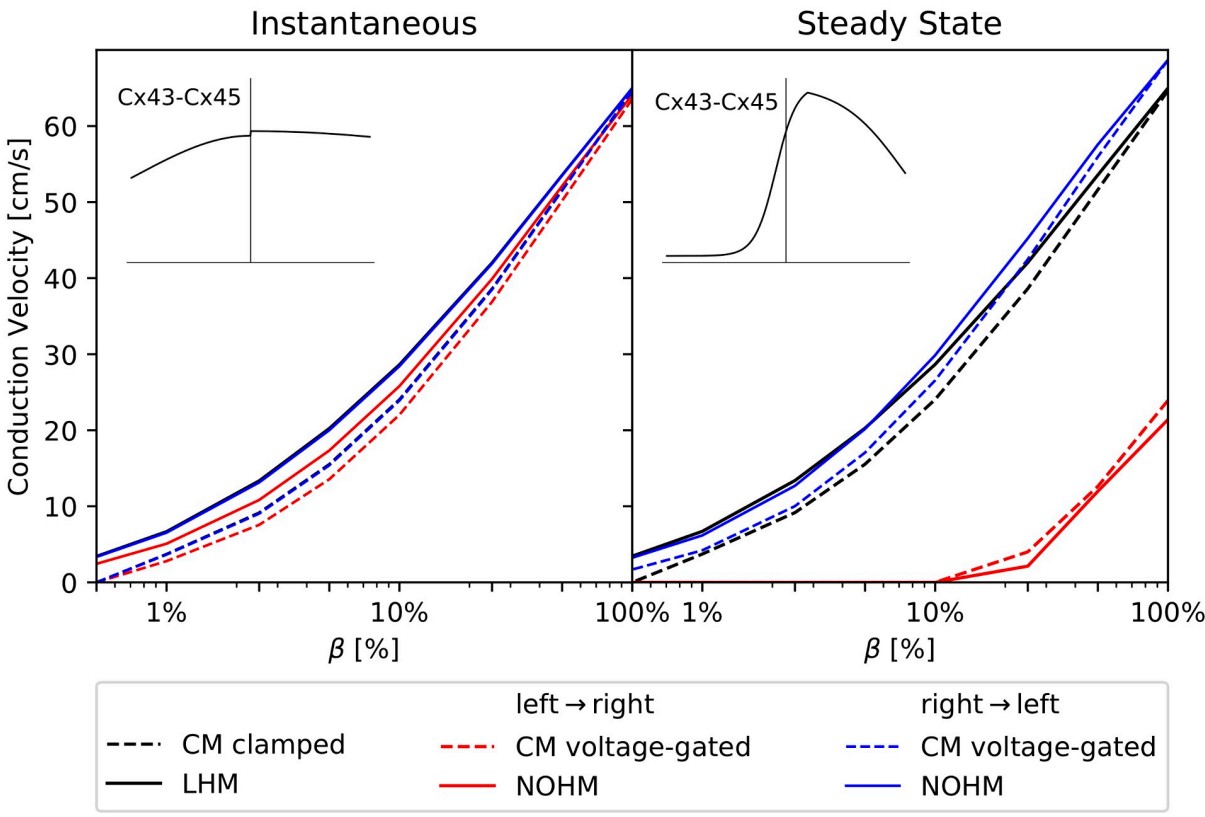

**Fig 6. Conduction velocity studies on a cardiac strand and the effect of gap-junction coupling for the Cx43-Cx45 channel with non-symmetric conductance distribution: (left) instantaneous conductance case, (right) steady-state conductance case.** Black color denotes voltage-independent models, red and blue colors denote voltage-dependent models. Predictions from gap-junction voltage-independent models CM clamped, and LHM were insensitive to the direction of wave propagation, whereas voltage-dependent models resulted in CVs that strongly depended on the direction of wave propagation for the steady-state conductance case.

sizes. An attenuated yet considerable mesh dependence was also observed for the LHM and NOHM.

## Discussion

In this article, we study the gap-junction-mediated electrical conduction in excitable cardiac tissue through a novel non-ohmic multiscale model. A unique feature of the proposed model is that tissue-level spatial conduction is fully informed by sub-cellular communication mechanisms, specifically by cytoplasmic and gap-junctional conductances. While the upscaling of conduction properties in excitable media has been the subject of some studies in the past using a linear homogenization theory approach [11, 12], our work offers a rigorous mathematical framework that delivers an effective non-linear model of conduction able to represent, at the tissue level, the non-Ohmic conduction that takes place at the sub-cellular level. Although our focus has been on understanding gap-mediated communication between cardiac myocytes, the present model of conduction can be extended to study the electrical propagation phenomena in other areas of biology, such as the neurosciences, where electrical synapsis occurring in the brain is highly regulated by neural gap junctions [31].

To validate the predictions and understand the unique features of the NOHM, we considered cellular models with and without voltage dependence at the gap junctions (CM clamped and CM voltage-gated) as well as a linear continuum model of conduction (LHM). The

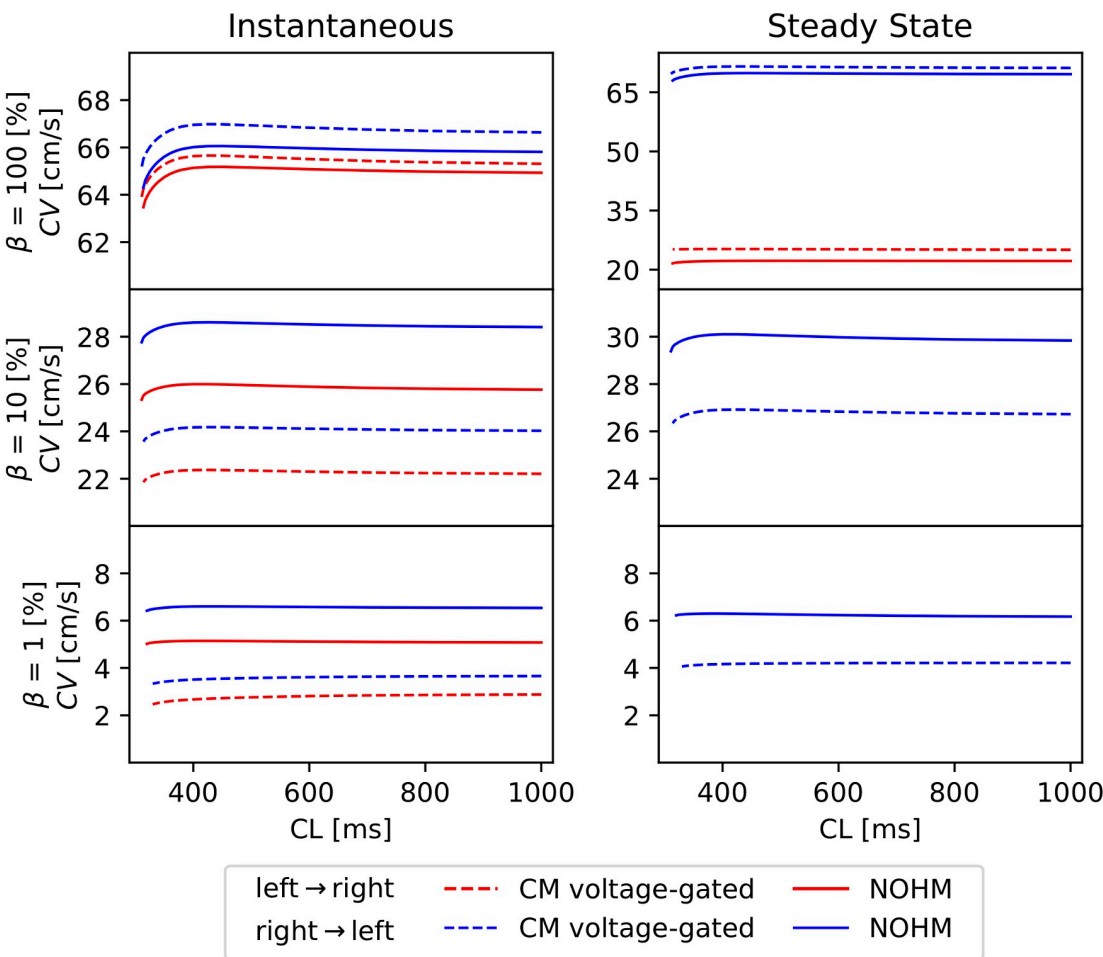

**Fig 7. Conduction-velocity restitution curves for the homomeric-heterotypic channel Cx43-Cx45 for high, low and very low GJc: (left) instantaneous conductance case, (right) steady-state conductance case.** CL = cycle length.

behavior of all four models of conduction was studied in the propagation of waves in a cardiac strand [24] with decreasing levels of GJc, see Fig 4. Features that arise in propagating action potentials under decreasing levels of coupling such as a steeper upstroke and a notch in the upstroke [5] are predicted both by the LHM and NOHM. Remarkably, this prediction is achieved at a fraction of the computational complexity involved in cellular models, as the number of degrees of freedom in continuum models are one order of magnitude smaller. An alternative approach is the use of hybrid multiscale models [32], which adaptively partition the domain to solve macroscopic cable equations in regions with low potential gradients and impose microscopic equations of conduction in regions of high potential gradients. While hybrid models can reduce the computational complexity of simulations, they involve a significant increase in the number of degrees of freedom when compared to standard homogenized models. We believe that the NOHM model offers the advantage of delivering accurate predictions while maintaining the computational cost similar to that of standard macroscopic continuum models. The balance between predictive power and computational cost remains one of the main hurdles in the development of patient-specific whole-heart simulations [33], which highlights the importance of developing accurate yet efficient tissue-level models.

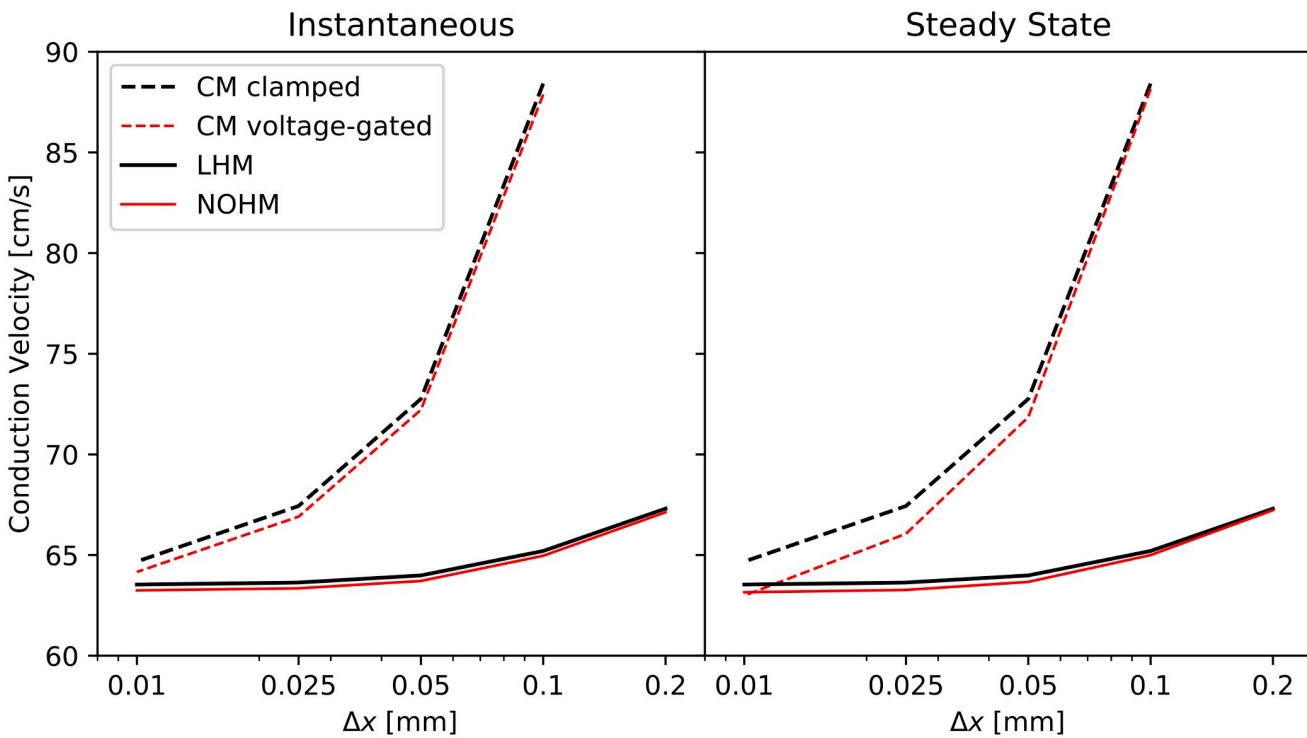

**Fig 8. The effect of spatial discretization of the conduction velocity: (left) instantaneous conductance, (right) steady-state conductance.** The conduction velocity in cellular models exhibit a stronger dependence on the mesh size than the continuum models of conduction.

The accuracy of the NOHM and LHM in predicting their cellular counterparts CM clamped and CM voltage-gated was assessed by studying the CV for a wide range of gap-junctional coupling levels for channels with symmetric conductance distributions, see Fig 5. In our work, we considered two limits of the dynamic conductance of gap junctions: the instantaneous and steady-state conductance cases. In the case of instantaneous conductance, the CV predicted by the NOHM was in general higher than the CV predicted by the CM voltage-gated, which was observed to decrease as the GJc was reduced [27]. A similar trend was observed for the LHM when compared with the CM clamped. Further, the NOHM and LHM resulted in similar CV curves. Previous studies have confirmed that the accuracy of LHM in predicting cardiac conduction, as dictated by CM clamped, consistently deteriorates as the GJc is decreased to low levels [13, 16]. Interestingly, for the case of steady-state conduction, substantial differences arise between voltage-dependent (NOHM, CM voltage-gated) and voltage-independent (LHM, CM clamped) models, with the former resulting in considerably lower CVs, see Fig 5 (left column). These results can be explained by noting the shape of the conductance distributions associated to the instantaneous and steady-state cases. In particular, the instantaneous conductance of the Cx43-Cx43 channel displays a flat shape with small variations within the range of transjunctional voltages (Fig 3, left), which resembles an Ohmic electrical response. Thus, in this case, the NOHM is not expected to differ from the LHM, as the conduction in both cases is fairly Ohmic, a behavior observed in our experiments (Fig 5, top left). In contrast, the steady-state conductance distribution for the Cx45-Cx45 channels presents a narrow bell shape (Fig 3, center), which is representative of a marked non-Ohmic electrical behavior. Notably, simulations associated with that conductance distribution are the ones that deliver the most different CV curves when the voltage dependence is included (Fig 5,

bottom right). These results confirm the ability of the NOHM to accurately upscale the voltage-dependent behavior of gap junctions, a feature not offered by standard homogenization models of conduction, such the LHM. Further, we note here that the difference in CV between cellular models CM clamped and CM voltage-gated for low GJc has been previously reported in the literature [5, 27], highlighting the importance of modeling the dynamic conductance of gap junctions for cases of low GJc. Decreased GJc takes particular relevance in the study of cardiac disease, as the reduction of gap-junctional communication has been correlated to a marked decreased of CV [34], and slow conduction is considered one of the main mechanisms of sustained reentrant arrhythmias [1, 35].

A unique feature of the NOHM model is its ability to upscale the particular features of voltage-dependent conduction mediated by homotypic and heterotypic combinations of homomeric connexons. In our simulations for the steady-state regime, action potentials resulting from channels composed by homotypic Cx43-Cx43 resulted in a considerably higher CV when compared to simulations considering homotypic Cx45-Cx45 channels (Fig 5 right column). This result is consistent with observations from dual whole-cell patch clamp experiments, where the lower CV in Cx45-Cx45 channels is explained by the higher sensitivity of conductance to transjunctional voltage [36]. We note, however, that for the instantaneous regime, the CV is more sensitive to the number of operational channels and unitary conductance than to gap junction voltage dependence. Further, here we showed that the asymmetry of conductance distribution found in heterotypic channels results in propagating action potentials whose CV strongly depends on the direction of propagation (Fig 6). The NOHM successfully captured this orientation dependence, as well as it was able to capture the conduction block predicted by the CM voltage-gated for low and very low GJc in the steady-state conduction regime. We also studied the effect of pacing rate for the Cx43-Cx45 channel, where both voltage-sensitive models resulted in similar restitution curves with higher CVs for the case of retrograde propagation (Fig 7). The orientation-dependent conduction, together with connexin coexpression, may partly explain the differences in CV for normal and retrograde conduction that have been observed in the sinoatrial node [37]. It is important to note that voltage-independent models of conduction (CM clamped, LHM) cannot capture this orientation dependence, as well as they fail to predict conduction block, resulting in a considerable overestimation of the CV for the case of normal wave propagation. Future developments should focus on combining the gap-junction conductance distributions, as several connexin types are typically co-expressed in cardiac tissue.

We assessed the effect of spatial discretization for the continuum and cellular conduction models considered in this study. Previous studies have shown that the numerical solution of continuum electrophysiology models depends on the level of spatial discretization [38, 39]. An interesting finding of this work is that the mesh dependence found in continuum models is accentuated for cellular models and that the consideration of voltage sensitivity in the conduction model does not strongly affect the relation between CV and mesh size (Fig 8). Future developments of numerical methods for the NOHM may include the consideration of enhanced spatial interpolation and temporal integration schemes [40, 41] which have shown to attenuate the mesh dependence of continuum model of cardiac propagation, allowing for the efficient simulation of larger domains.

Our current work can be extended in several directions. First, the theoretical framework for the NOHM model should be extended to consider the 3D case of cardiac conduction, including the case of anisotropic conduction typically observed in cardiac tissue. We include a heuristic derivation in Remark 2 of the S1 Appendix that points towards this direction. Another important limitation is the consideration of the transmembrane potential, instead of the intra- and extra-cellular potentials, in modeling intercellular conduction and ionic ionic currents. In

particular, (3) assumes that transjunctional voltage is the jump in transmembrane potential rather than the jump in intra-cellular potential. We note that such consideration is valid when the extra-cellular potential is constant, an assumption that can be debated in more general contexts of conduction. Future contributions should revisit this assumption by explicitly modeling the extra-cellular potential, i.e., considering bidomain formulations of the continuum electrophysiology problem [9]. Second, we note that the time dependence of gap-junction gating that dynamically modulates the conduction has not been considered in this work. Such a dynamic effect has shown to strongly modulate the conductance response of gap junctions [28, 29]. To date, the time-dependent gating of gap junctions has been incorporated in a few cellular models of cardiac conduction [5, 27], showing the importance of both voltage- and time-dependent dynamics. We note here that it takes several seconds for a gap-junction channel to reach steady-state conductance. Such a time scale can be much longer than the time window where transjunctional voltage is large during normal conduction, i.e., during action potential upstroke. Thus, during normal conduction, the steady-state regime is not expected to occur. In contrast, under cases of poor intercellular coupling, large transjunctional voltage can occur for longer periods, which potentially drive the gap-junction towards a steady-state conduction regime [27]. In this work, we only considered two limiting regimes of the dynamic conductance which yield very different behavior as the steady-state regime typically displays conductance distributions that are more sensitive to transjunctional voltage than those found for the instantaneous-conductance regime. Thus, an interesting avenue of research is the development of multiscale formulations of cardiac tissue conduction that incorporate time-dependent gating dynamics of gap junctions. Third, intercellular communication mechanisms other than gap junctions should be integrated into this theoretical framework. Sodium channels have been reported to co-localize with gap junctions at the intercalated discs, creating an ephatic coupling effect that has been associated to conduction during gap-junction blockage [35]. Further, the spatial distribution of sodium channels around the cellular membrane and on the intercalated discs has been studied using detailed cell-to-cell computational simulations to conclude that channel spatial distribution strongly affect the cardiac conduction [42]. Since the ephatic effect has been considered in homogenization schemes of cardiac conduction in the past by including a cleft-to-ground resistance in the microscopic model of conduction [13, 32], we foresee that future versions of the NOHM could equally incorporate this effect, potentially in 3D formulations with non-uniform distributions of channels. Finally, the applicability of the NOHM model should be tested in the simulation of conduction in the whole heart during diseased conditions [33].

## Supporting information

**S1 Appendix. Formulation details.** Details and proofs for the asymptotic formulation and the numerical solution for the NOHM model are presented here.
(PDF)

## Author Contributions

**Conceptualization:** Daniel E. Hurtado, Grigory Panasenko.

**Formal analysis:** Daniel E. Hurtado, Javiera Jilberto, Grigory Panasenko.

**Funding acquisition:** Daniel E. Hurtado.

**Investigation:** Daniel E. Hurtado, Javiera Jilberto.

**Methodology:** Daniel E. Hurtado, Javiera Jilberto.

**Project administration:** Daniel E. Hurtado.

**Resources:** Daniel E. Hurtado.

**Software:** Javiera Jilberto.

**Supervision:** Daniel E. Hurtado.

**Validation:** Daniel E. Hurtado, Javiera Jilberto.

**Visualization:** Daniel E. Hurtado, Javiera Jilberto.

**Writing – original draft:** Daniel E. Hurtado, Javiera Jilberto, Grigory Panasenko.

**Writing – review & editing:** Daniel E. Hurtado.

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
