## [Decision Letter · Decision Letter 0]

1 Aug 2019

Dear Dr Hurtado,

Thank you very much for submitting your manuscript 'Non-ohmic tissue conduction in cardiac electrophysiology: upscaling the non-linear voltage-dependent conductance of gap junctions' for review by PLOS Computational Biology. Your manuscript has been fully evaluated by the PLOS Computational Biology editorial team and in this case also by independent peer reviewers. The reviewers appreciated the attention to an important problem, but raised some very substantial concerns about the manuscript as it currently stands. While your manuscript cannot be accepted in its present form, we are willing to consider a revised version in which the issues raised by the reviewers have been adequately addressed. We cannot, of course, promise publication at that time.

Sincerely,

Aslak Tveito

Guest Editor

PLOS Computational Biology

Daniel Beard

Deputy Editor

PLOS Computational Biology

[LINK]

Reviewer's Responses to Questions

**Comments to the Authors:**

Reviewer #1: The study by Hurtado et al presents a cardiac tissue model that accounts for nonlinearity of gap junction conductance, specifically deriving and presenting the numerical solution of a homogenized model that incorporates voltage-dependent gap junctions. As noted by the authors, non-ohmic dynamics is an under-appreciated aspect of gap junction behavior that is typically not accounted for in many cardiac tissue models. This study presents a nice potential approach to account for these details without a significant increase in computational complexity.

However, there are several major issues for the authors to address:

Major:

1. The biophysical basis for the non-Ohmic behavior is a consequence of the voltage-dependent gating of the gap junction hemichannels (similar to the voltage-dependent gating of other sarcolemmal ion channels). My most significant concern is that the model formulation appears to neglect a critical aspect of the gating behavior, in particular that the gap junction conductance, in addition to being a function of transjunctional voltage (Vj), is also time-dependent. That is, the gating of the gap junction protein hemichannels has a time-dependence that also depends on Vj. This is demonstrated in a wide range of studies, see for example work from Weingart (including ref 3), Veenstra, Bukauskas, Bennett, and many others.

The time constant for changes in gap junction conductance is generally found to be a decaying function of the Vj magnitude, with values on the order of a few seconds when Vj = 0 mV. As presented, the model formulation here appears to assume that the gating of the gap junctions is instantaneous. This is problematic because, in the model, gap junction conductance changes are thus much faster than the dynamics of sarcolemmal ion channels, whereas physiologically, gap junction conductance changes are much slower.

When cells are well coupled, the absolute value of Vj is generally not much greater than 0 mV for durations on the order of at most 10s of milliseconds, and thus gap junction conductance generally does not approach the reduced steady-state levels shown at the extreme Vj values in Figure 3.

However, when cells are poorly coupled, i.e., low baseline gap junction conductance levels, large Vj values can occur for longer durations, which in turn does result in transient decreases in gap junction conductance. This has been previously demonstrated by Henriquez et al (ref 20 in the manuscript), and more recently by Weinberg (Chaos, 2017). At a minimum, the authors should compare their work with these prior studies, which are two of the few studies that have accounted for non-linear gap junction conductance in a tissue model, with Weinberg also including electrical field coupling.

The lack of accounting for the time-dependence of gap junction conductance changes is a significant limitation that detracts from potential impact of the study. However, perhaps this can be incorporated into the proposed framework by including additional gating variables into the w gating variable vector, following approaches similar to either the Henriquez et al or Weinberg studies noted above, which both include time-dependent gap junction conductance changes.

2. My other significant concern is the lack of description of the cell-chain model from Kucera et al (ref 5) that is considered the “baseline” in this study. The paper by Kucera and colleagues described two variants of their model, so it is not clear which version of the Kucera et al model is used and what are the associated parameters. In particular, in addition to discretizing each cell into membrane patches of 10 um (as the authors note), Kucera et al also describe a “non-cleft” and “cleft” version of the model, in which electric field coupling occurs via extracellular current in the intercellular cleft in the “cleft” version. Additionally, Kucera et al study the significance of redistributing the voltage-gated sodium current from axial to intercalated disk membrane patches and variations in intercellular cleft width. (The authors note this redistribution of sodium channels in the Discussion but not in the description of the baseline model.)

Since the focus of the Kucera et al paper is the “cleft” version, one would assume that this version of the model was used in the current study, but this needs to be clarified. Regardless of which version of the Kucera et al model is used, both versions of the model include a “Rgap” term – a constant gap junction resistance between cells. It is not clear if the non-ohmic conductance is also incorporated into the baseline model, or if Rgap is a constant in the baseline model.

Similarly, the linear homogenized model that is compared (LHM) from Hand and Peskin (ref 12) also incorporates intercellular cleft electric field coupling and sodium channel distributions. However, the authors describe this model as a “standard cable model.” This is a confusing description, because the classical description of the cable model or the monodomain model does not include electric field coupling and assumes uniform distribution of sodium channels. A significantly more detailed description of these models used for comparison is needed.

3. The authors should expand significantly on the reasons why the non-ohmic model and the baseline model agree in some parameter regimes (specifically higher coupling) and disagree in other regimes (weaker coupling). In the regimes where the models disagree, then presumably there are some assumptions of the derivation that fail, such that the homogenization is not valid. These are important limitations that the authors should comment on and discuss, especially since, as the authors note, that slower conduction is often pro-arrhythmic and is thus of significant interest in simulations.

4. What is the baseline gap junction conductance value associated with the simulations in Figure 4? Based on the conduction velocity values, it appears the cells are well coupled, similar to as in Figure 1A.

In this well-coupled case, a conduction velocity of 50 cm/s implies that propagation of a distance of 100 um will take 0.2 ms, which is less than the duration of the cardiac action potential upstroke, so Vj magnitudes are probably on the order of 10-20 mV at most. Even without accounting for the time-dependence of the gap junction conductance as noted in comment 1, based on the curves in Figure 3, steady-state conductance levels are within 20% of the baseline value, so it is surprising that conduction velocity values differ by nearly 50%. Can the authors explain this result by examining the Vj curves and associated changes in gap junction conductance along the cable?

5. The differences in conduction velocity for different directions shown for the heterotypic gap junction, illustrated in Fig 4C, is one of the more interesting results of the paper. However, this point is demonstrated for a single case (i.e., one unknown value of gap junction coupling, see previous comment), and thus it is not clear for what conditions these directional differences are small or large. For example, are there conditions in which propagation fails in one direction but not the other? The authors should show a plot similar to Fig 2A plotting conduction velocity for both directions for different conductance levels.

It would also be interesting to study if there is a pacing rate dependence. For example, are there conditions in which conduction in both directions is similar at slow pacing rates, but differs for faster pacing rates?

6. The derivation of the model shown in the Appendix is fairly difficult to follow. An important contribution would be specifically highlighting how this derivation differs from the homogenization required for such a model in which gap junction conductance is constant.

Minor:

1. The sentence beginning with “Alternatively, …” at line 48 is an incomplete sentence.

2. Line 67, “an” should be “and”

3. In Fig. 4, the conduction velocity for the Cx43-Cx45 gap junction model is given as 32.1 cm/s in panel A and then 32.2 cm/s in panel B. Is this a typo since – as I understand it – these are referring to the same simulation condition?

Reviewer #2: The paper describes the development of a macroscopic tissue model for electrical conduction in cardiac tissue, which incorporates non-linear voltage-dependent conduction through gap junctions. The topic is important and relevant for the research community in computational cardiac electrophysiology, and the paper presents a new modeling approach that could potentially have significant implications. However, I have some concerns related to the model derivation and the discussion of the results, which should be improved before publication. Furthermore, although the manuscript is generally well written, the overall structure and ordering (Results-Discussion-Methods) makes it somewhat hard to read. I assume the structure is dictated by the journal, which raises the question of whether PLOS Computational Biology is the best target for this fairly mathematical and model-oriented manuscript.

Major concerns:

1. The model derivation described in the Methods section is not based on physically meaningful properties. In appendix S1 the homogenization is performed in terms a generic microscopic potential and microscopic current density, which in this context must be interpreted as intracellular properties. However, in eq (3) in the Methods section the intracellular/GJ current density is computed by multiplying the transmembrane potential with the cytoplasm/GJ conductivity. This is only correct if the extracellular potential is constant. If this is assumed it should be mentioned explicitly in the derivation, since it is a significant limitation with potential implications for the model’s range of validity. I would recommend that the model derivation is based explicitly on balance of intra- and extracellular currents, expressed in terms of intra- and extracellular potentials, and that all assumptions leading to the final model are made explicit. It should also be considered if a more generic 2D/3D version of the model could be derived, since the restriction to 1D is a severe limitation.

2. The discussion of the results in relation to existing models is very limited. The GJ conduction models used by the authors seem well justified, and show interesting (although not entirely surprising) effects on conduction velocity. However, there are several alternative formulations of GJ conductance, including a variety of non-Ohmic and voltage-gated formulations. A comprehensive review of all existing models is obviously beyond the scope of the paper, but I would like to see a more thorough discussion of the results in the context of existing literature.

3. The baseline model is very briefly described. Although this model is described in some detail in the cited reference [5], it would be useful to recapitulate the main equations of the model in the present manuscript, or in a supplement. This would make the similarities and differences between the two models more apparent, and highlight relations between the model’s parameters. In particular, it is not clear whether non-Ohmic GJ conduction is used for the baseline model, or if the original Ohmic formulation from [5] is used. Furthermore, it would be interesting to see the effect of discretization parameters both for the baseline model and the homogenized model. Does the conduction velocity of the baseline model change if the number of nodes per cell is increased or reduced? And what about the discretization of the NOM and LHM models?

Minor issues:

- One page 3, lines 16-20, the discussion of existing literature could be more precise. The main topic of reference [5] is the study of sodium channel distribution related to GJs, which is not addressed in the present paper, and not to the GJ conduction itself.

- Page 3, lines 30-34: The formulation suggests that the monodomain model is based on the assumption of isotropic conductivity, which is not the case. Also, the most relevant model to reference in this context would be the bidomain model, since this is considered the most accurate model of cardiac electrophysiology, but is also based on the Ohmic assumptions used for the cable equation.

- Page 4, lines 48-51: The sentence is incomplete.

- Page 4, lines 63-65: I assume the current referred to is a transmembrane current, but it would be useful to make this explicit.

- Page 5, line 68: Is the LHM model the same model that would be obtained from inserting Ohmic GJc in the homogenization applied in this paper? If so, it could be useful to formulate it in this way, to make the model formulation and parameter specification more precise.

- On page 5, lines 74-79, it would improve readability if the change of GJc was explicitly referring to model parameter, stating which parameters are changed in the three models (LHM, NOM, baseline).

- Page 8, lines 142-143: Although capturing the low-conductance behavior is a strong feature of the proposed model, I would not describe the result as “remarkable”. As far as I can tell, all the proposed GJc models tested in the paper effectively shut down conduction as the voltage difference becomes large. Since low GJc will lead to increased cell-to-cell voltage difference, it is quite intuitive that the proposed models give conduction slowing compared with an Ohmic model. This could be commented on in the discussion.

- Page 10, eq (4): Why is the voltage jump divided by the cell length? (And there seems to be a mix of subscripts j and k)

- Page 11, line 211: Why is the ionic current a mapping from (R x R) to R?

- Page 11, line 227: The authors are to be applauded for intending to make all codes available for download. However, the listed github-repository is empty.

Reviewer #3: Review is uploaded

**Have all data underlying the figures and results presented in the manuscript been provided?**

Reviewer #1: No: The methods used to derive much of the figure results are unclear. See Author Comments.

Reviewer #2: None

Reviewer #3: None

PLOS authors have the option to publish the peer review history of their article (what does this mean?). If published, this will include your full peer review and any attached files.

Reviewer #1: No

Reviewer #2: No

Reviewer #3: No

---

## [Decision Letter · Decision Letter 1]

23 Dec 2019

Dear Dr Hurtado,

Thank you very much for submitting your manuscript, 'Non-ohmic tissue conduction in cardiac electrophysiology: upscaling the non-linear voltage-dependent conductance of gap junctions', to PLOS Computational Biology. As with all papers submitted to the journal, yours was fully evaluated by the PLOS Computational Biology editorial team, and in this case, by independent peer reviewers. The reviewers appreciated the attention to an important topic but identified some aspects of the manuscript that should be improved.

We would therefore like to ask you to modify the manuscript according to the review recommendations before we can consider your manuscript for acceptance. Your revisions should address the specific points made by each reviewer and we encourage you to respond to particular issues Please note while forming your response, if your article is accepted, you may have the opportunity to make the peer review history publicly available. The record will include editor decision letters (with reviews) and your responses to reviewer comments. If eligible, we will contact you to opt in or out.raised.

- Supporting Information uploaded as separate files, titled 'Dataset', 'Figure', 'Table', 'Text', 'Protocol', 'Audio', or 'Video'.

We hope to receive your revised manuscript within the next 30 days. If you anticipate any delay in its return, we ask that you let us know the expected resubmission date by email at ploscompbiol@plos.org.

Two of the three reviewers are now satisfied with your manuscript, and the third ask for clarification of some technical issues that I hope you will be able to address. 

Sincerely,

Aslak Tveito

Guest Editor

PLOS Computational Biology

Daniel Beard

Deputy Editor

PLOS Computational Biology

[LINK]

Reviewer's Responses to Questions

**Comments to the Authors:**

Reviewer #1: The authors have mostly addressed my concerns. However, there are still a few issues that I would like the authors to address.

1. The inclusion of the instantaneous gap junction gating studies are a nice addition to the manuscript. The authors should more clearly describe the differences between the instantaneous and steady-state conductance levels. The authors simply state “Given the time-dependent behavior of the conductance of GJs …” This explanation is not sufficient for the typical reader of the journal.

Related to this, while the authors have made efforts to more clearly highlight the time dependence of gap junctional gating, there is still no mention of the time scale for this gating process, specifically that the time to reach steady state can be on the order of seconds, which is longer than the cardiac action potential and certainly longer than the action potential upstroke. The manuscript highlights the significant differences between results when assuming instantaneous vs steady-state gap junction conductance levels; however the manuscript needs to also clearly highlight that steady-state values are only likely reached in cases of poor coupling when large magnitude Vj values can persistent for more than milliseconds.

2. It is still not clear to me why the paper from Kucera, Rohr, and Rudy (ref 21) is highlighted as cellular model for comparison. As previously commented, this paper describes two model versions, the “non-cleft” and “cleft” version, with the non-cleft version serving as the control or comparison. The focus of that paper is on the cleft version that considers preferential localization of sodium channels and electric field or ephaptic coupling in the intercellular cleft. At a minimum, the authors should describe the cellular model as the “non-cleft” model from Kucera, Rohr, and Rudy, which is identical to the model illustrated in Fig. 2 in the revised manuscript.

However, this paper is far from the first to investigate discontinuous propagation in cardiac tissue and discretize the cell into multiple compartment, as in the non-cleft model. There are several earlier papers from Plonsey, Henriquez, and Rudy (and likely many others) using similar models. Diaz, Rudy, and Plonsey, Annals of Biomed Eng, 1983; Henriquez and Plonsey, Med & Biol. Eng. & Comput, 1987; and Shaw and Rudy, Circ Res, 1997 are three such examples.

3. The instantaneous gap junction conductance for the Cx43-Cx43 and Cx45-Cx45 pairs shown in Figure 3 do not appear to be symmetric with respect to Vj (although it is somewhat difficult to tell this by eye). Are these relationships asymmetric and if so, why? It is not obvious why the directionality should matter in homotypic channels.

4. Is the shortest CL values in the CV restitution curves in Fig. 7 the shortest CL that elicited propagation in each case, or do all curves run to some value around 300 ms? If the later, simulations should be run for CL values down to the loss of propagation or capture for each case. This will demonstrate if there are differences between loss of propagation between the different cases. The CV restitution curves are typically much steeper for these short CL values, and thus there are likely to be much greater differences between the different cases. This is particular relevant to simulation of tachyarrhymias.

Minor:

1. For the results shown in Figures 4-6 and 8, what is the pacing cycle length used? This should be included in either the text or figure captions.

Reviewer #2: The authors have addressed all the concerns I had with the original manuscript.

Reviewer #3: The revision is satisfactory.

**Have all data underlying the figures and results presented in the manuscript been provided?**

Reviewer #1: Yes

Reviewer #2: None

Reviewer #3: None

PLOS authors have the option to publish the peer review history of their article (what does this mean?). If published, this will include your full peer review and any attached files.

Reviewer #1: No

Reviewer #2: Yes: Joakim Sundnes

Reviewer #3: No

---

## [Decision Letter · Decision Letter 2]

15 Jan 2020

Dear Dr Hurtado,

We are pleased to inform you that your manuscript 'Non-ohmic tissue conduction in cardiac electrophysiology: upscaling the non-linear voltage-dependent conductance of gap junctions' has been provisionally accepted for publication in PLOS Computational Biology.

In the meantime, please log into Editorial Manager at https://www.editorialmanager.com/pcompbiol/, click the "Update My Information" link at the top of the page, and update your user information to ensure an efficient production and billing process.

One of the goals of PLOS is to make science accessible to educators and the public. PLOS staff issue occasional press releases and make early versions of PLOS Computational Biology articles available to science writers and journalists. PLOS staff also collaborate with Communication and Public Information Offices and would be happy to work with the relevant people at your institution or funding agency. If your institution or funding agency is interested in promoting your findings, please ask them to coordinate their releases with PLOS (contact ploscompbiol@plos.org).

Thank you again for supporting Open Access publishing. We look forward to publishing your paper in PLOS Computational Biology.

Sincerely,

Aslak Tveito

Guest Editor

PLOS Computational Biology

Daniel Beard

Deputy Editor

PLOS Computational Biology

Reviewer's Responses to Questions

**Comments to the Authors:**

Reviewer #1: The authors have addressed my concerns. Thank you. Note there is a Figure referencing typo on page 19, line 300.

**Have all data underlying the figures and results presented in the manuscript been provided?**

Reviewer #1: Yes

PLOS authors have the option to publish the peer review history of their article (what does this mean?). If published, this will include your full peer review and any attached files.

Reviewer #1: No

---

## [Editor Report · Acceptance letter]

13 Feb 2020

PCOMPBIOL-D-19-01074R2 

Non-ohmic tissue conduction in cardiac electrophysiology: upscaling the non-linear voltage-dependent conductance of gap junctions

Dear Dr Hurtado,

I am pleased to inform you that your manuscript has been formally accepted for publication in PLOS Computational Biology. Your manuscript is now with our production department and you will be notified of the publication date in due course.

With kind regards,

Sarah Hammond
